# A Review of Lipidomics of Cardiovascular Disease Highlights the Importance of Isolating Lipoproteins

**DOI:** 10.3390/metabo10040163

**Published:** 2020-04-23

**Authors:** Ming Ding, Kathryn M. Rexrode

**Affiliations:** 1Department of Nutrition, Harvard School of Public Health, Boston, MA 02115, USA; 2Division of Women’s Health, Brigham and Women’s Hospital, Harvard Medical School, Boston, MA 02115, USA; KREXRODE@bwh.harvard.edu

**Keywords:** lipidomics, cardiovascular disease, lipoproteins, HDL and LDL

## Abstract

Cutting-edge lipidomic profiling measures hundreds or even thousands of lipids in plasma and is increasingly used to investigate mechanisms of cardiovascular disease (CVD). In this review, we introduce lipidomic techniques, describe distributions of lipids across lipoproteins, and summarize findings on the association of lipids with CVD based on lipidomics. The main findings of 16 cohort studies were that, independent of total and high-density lipoprotein cholesterol (HDL-c), ceramides (d18:1/16:0, d18:1/18:0, and d18:1/24:1) and phosphatidylcholines (PCs) containing saturated and monounsaturated fatty acyl chains are positively associated with risks of CVD outcomes, while PCs containing polyunsaturated fatty acyl chains (PUFA) are inversely associated with risks of CVD outcomes. Lysophosphatidylcholines (LPCs) may be positively associated with risks of CVD outcomes. Interestingly, the distributions of the identified lipids vary across lipoproteins: LPCs are primarily contained in HDLs, ceramides are mainly contained in low-density lipoproteins (LDLs), and PCs are distributed in both HDLs and LDLs. Thus, the potential mechanism behind previous findings may be related to the effect of the identified lipids on the biological functions of HDLs and LDLs. Only eight studies on the lipidomics of HDL and non-HDL particles and CVD outcomes have been conducted, which showed that higher triglycerides (TAGs), lower PUFA, lower phospholipids, and lower sphingomyelin content in HDLs might be associated with a higher risk of coronary heart disease (CHD). However, the generalizability of these studies is a major concern, given that they used case–control or cross-sectional designs in hospital settings, included a very small number of participants, and did not correct for multiple testing or adjust for blood lipids such as HDL-c, low-density lipoprotein cholesterol (LDL-c), or TAGs. Overall, findings from the literature highlight the importance of research on lipidomics of lipoproteins to enhance our understanding of the mechanism of the association between the identified lipids and the risk of CVD and allow the identification of novel lipid biomarkers in HDLs and LDLs, independent of HDL-c and LDL-c. Lipidomic techniques show the feasibility of this exciting research direction, and the lack of high-quality epidemiological studies warrants well-designed prospective cohort studies.

## 1. Introduction

Cardiovascular disease (CVD) is the leading cause of death globally, accounting for 17.8 million deaths per year [1]. Thus, early prevention and effective treatment significantly impact public health. Plasma lipid biomarkers including high-density lipoprotein cholesterol (HDL-c), low-density lipoprotein cholesterol (LDL-c), and triglycerides (TAG) have been used to assess the risk of CVD for decades [2,3,4]. HDL-c and LDL-c, together with other components of the Framingham heart score, predicted approximately 75% of CVD risk [5]. These lipid biomarkers are clinically used for the evaluation of CVD risk and decision on CVD treatment [6]. 

Emerging lipidomic techniques allow for high-throughput profiling of thousands of lipids categorized into five main species, namely, glycerolipids, phospholipids, sphingolipids, cholesterols, and free fatty acids (FFA). The development of the lipidomics field is particularly relevant to understanding the mechanisms of CVD, as lipids have been shown to play a key role in the pathophysiology of CVD. Thus, lipidomics can be incorporated into CVD epidemiology to enhance our understanding of how lipids (i.e., individual lipids and fatty acyl chains esterified with the glycerol backbone) affect the risk of CVD and potentially improve CVD prediction in addition to HDL-c and LDL-c. In this review, we introduce lipidomic techniques, summarize recent advances in lipidomics of CVD, compare lipid composition across lipoproteins, and highlight areas of future research based on the existing literature. 

## 2. Lipidomics Techniques

### 2.1. Liquid Chromatography (LC)-Based Techniques

LC–mass spectrometry (LC–MS) is one of the popular methods for lipidomics measurement because of the relatively low cost and high sensitivity of lipid measurement. LC–MS begins with the extraction of lipids from plasma. The most popular method is liquid–liquid extraction using a mixture of dichloromethane/methanol or butanol/methanol [7,8,9]. Methanol destroys and precipitates lipoproteins, and dichloromethane guarantees the effective extraction of a wide range of lipid species from the precipitated lipoproteins. As methanol precipitates lipoproteins, lipids in total plasma, rather than within lipoproteins, are measured, which is a particular feature of the LC-based technique. Thus, in order to conduct lipidomics of lipoproteins, lipoproteins must be first isolated prior to LC-based measurements; density-gradient ultracentrifugation (UC) represents the gold standard method for the isolation and quantification of HDL and LDL cholesterol [10].

LC separates lipids based on their physicochemical properties, i.e., polar head-group classes, carbon-chain length, and the number of double bonds, as indicated by the retention time. LC separation includes normal-phase LC, with a polar stationary phase and a non-polar mobile phase, and reversed-phase LC, with a non-polar stationary phase and a polar mobile phase. After chromatographical separation, the isolated lipids enter the ionization source and undergo ionization, and the produced lipid fragments are detected using a mass analyzer for structure identification. As LC separates and concentrates lipids simultaneously, one advantage of LC–MS is the ability to measure thousands of lipids with high sensitivity, while requiring a very small sample volume. However, due to the similarity of the separation times, one limitation of LC–MS is that it cannot detect lipid isomers that may play an important role in the development of CVD [11,12,13,14,15], including structural (i.e., trans and cis alkenes) and positional isomers (i.e., depending on the position of double bonds on the fatty acyl chain and on the stereospecific numbering (sn) position of the fatty acyl chain on the glycerol backbone). Moreover, another limitation of LC–MS is ion suppression, which is defined as reduced ionization, whereby reduced numbers of ions are available to reach the detector. Ion suppression decreases detection capability and measurement accuracy.

### 2.2. Shotgun Lipidomics

Shotgun lipidomics, a technique prevalent at the beginning of lipidomics research, directly infuses lipids into an electrospray ionization mass spectrometer (ESI/MS) for the detection of lipids. In contrast to LC-based methods that separate lipids chromatographically, shotgun lipidomics separates hundreds of lipids by electrospray ionization [16]. ESI uses electrospray under high voltage to produce multiple-charged ions [17], and therefore, shotgun lipidomics can quantify hundreds of lipids with relative simplicity of operation and short run times. However, it only allows the detection of abundant lipid species and thus has lower sensitivity than LC-based methods. Moreover, distinguishing lipid isomers is difficult when using shotgun lipidomics due to its low efficiency in separating aggregated lipids. For lipidomics of lipoproteins using shotgun techniques, the isolation of lipid fractions (HDL and LDL) is needed prior to ESI. Similar to LC–MS, ion suppression decreases the sensitivity of lipid identification and is a limitation of ESI. 

### 2.3. Nuclear Magnetic Resonance (NMR)

In recent years, NMR has increasingly been used in lipidomic assessment, and its functionality relies on the inherent nuclear spin of atomic nuclei, including ^1^H, ^13^C, ^15^N, and ^31^P, that generates a magnetic field associated with them. NMR stimulates the nuclear spin through rapid changes in an external magnetic field and then records the electromagnetic radiation released following nuclei relaxation. The resonance frequency of the energy released by the atomic nuclei reflects the microenvironment of adjacent nuclei. Thus, NMR has been widely used for structural elucidation. As the biological sample is physically isolated from the NMR instrument, NMR is nondestructive, and lipoproteins in plasma are preserved. As NMR is sensitive to the size and density of macromolecule aggregates, the density, size, and particle number of lipoprotein subclasses can be quantified using NMR [18]. NMR can also efficiently and accurately quantify total lipids and the content of lipid species (i.e., total TAG, total cholesterol, total phospholipids, and total sphingolipids) within lipoprotein subclasses [18]. However, given that lipoprotein are complex compounds composed of various lipids, the identification of individual lipids within lipoprotein subclasses is difficult using NMR. Moreover, compared to LC–MS and shotgun lipidomics, NMR-based lipidomics requires relatively long measurement times and has a low sample throughput.

## 3. Summary of Recent Studies on Lipidomics for CVD

We searched cohort-based studies on lipidomics and CVD outcomes. We identified 16 studies which are listed in Appendix A. As to methods of lipid measurement, two early studies used shotgun lipidomics [19,20], two studies used NMR measuring total lipids, triglycerides, and cholesterol within l4 lipoprotein subclasses [21,22], and the majority of studies used LC–MS, measuring hundreds of lipids [16,17,18,19,20,21,22,23,24,25,26,27]. Eight of the 16 studies were lipidomics/metabolomics-wide association studies that assessed associations of all measured lipids with CVD outcomes [19,20,21,22,23,24,25,26], six studies used a candidate-lipid approach by focusing on specific lipids such as ceramides [27,28,29,30,31,32], and two studies classified lipids into groups and examined global associations of lipid species with CVD outcomes [33,34]. Of the 16 studies, 6 studies included participants with a history of coronary heart disease (CHD) and focused on secondary prevention [23,24,26,28,29,31], and 10 studies focused on primary prevention of CVD [19,20,21,22,25,27,30,32,33,34]. Most of the included studies adjusted for lipid biomarkers such as total cholesterol and HDL-c as covariates in the model. All included studies were conducted in Caucasians, except for one study carried out in a Chinese population [22]. 

### 3.1. Main Findings 

Table 1 summarizes the main findings of the 16 studies with regard to lipid classes and CVD. First, ceramides, particularly Cer (d18:1/16:0), Cer (d18:1/18:0), and Cer (d18:1/24:1), were positively associated with the risk of CVD independent of HDL-c and LDL-c in multiple studies. The lipidomic-wide association LURIC study first showed that the three ceramides were positively associated with the risks of recurrent CHD and mortality and that Cer (d18:1/24:0) was inversely associated [23]; however, these associations were eliminated after adjusting for multiple testing. The ATHEROREMO-IVUS study measured ceramides that were identified in the LURIC study and showed that Cer (d18:1/16:0) and the Cer (d18:1/24:1)/Cer (d18:1/24:0) ratio were positively associated with major adverse cardiac events [28,29]. Thereafter, the FINRISK 2002 study [30], the Corogene study [31], and the PREDIMED trial [32] specifically focused on the four mentioned ceramides and consistently found that Cer (d18:1/16:0), Cer (d18:1/18:0), and Cer (d18:1/24:1) were positively associated with CVD outcomes. Moreover, the ADVANCE and LIPID trials conducted lipidomic-wide association studies and found that, out of 300 lipids, glucosylceramides (d18:1/16:0, d18:1/18:0, and d18:1/24:1) had a strong positive association with recurrent CVD and CVD mortality [24,26]. 

Second, independent of HDL-c and LDL-c, phosphatidylcholines (PCs) containing saturated fatty acyl (SFA) chains and monounsaturated fatty acyl (MUFA) chains were associated with a higher risk of CVD, while PCs containing polyunsaturated fatty acyl (PUFA) chains were inversely associated with it. In the PREDIMED trial, a lipid cluster including highly unsaturated PCs and cholesterol esters (CEs) was inversely associated with the risk of CVD [32,33,34]; however, these analyses did not adjust for total and HDL cholesterol. The WHI identified PCs with SFA and MUFA strongly associated with the risk of CHD, and the associations persisted after adjusting for total and HDL cholesterol [25]. Similarly, in analyses that adjusted for HDL-c and LDL-c, the LURIC study, ADVANCE trial, and LIPID study found that PCs containing PUFA, particularly the omega-6 fatty acyl chain (C20:4), were inversely associated with CVD outcomes, and PCs containing SFA and MUFA were positively associated with CVD outcomes [23,24,26]. 

Third, after adjusting for HDL-c and LDL-c, it was found that lysophosphatidylcholines (LPC) may be positively associated with the risk of CVD. The LIPID and ADVANCE trials identified positive associations with the risk of CVD for LPCs, including LPC (20:1), LPC (O-22:0), LPC (O-22:1), LPC (O-24:0), LPC (O-24:1), and LPC (O-24:2) [24,26]. However, the MDC study, TwinGene study, and LURIC study identified LPC (16:0), LPC (18:0), LPC (18:1), LPC (18:2), and LPC (20:4) inversely associated with the risk of CVD outcomes [19,23,27]. No associations of LPCs with the risk of CHD were documented in the WHI study [25]. 

Fourth, TAGs and CEs with SFA and MUFA chains were significantly associated with a high risk of CVD; however, these associations may be confounded by HDL-c and LDL-c. The Bruneck study found that TAGs and CEs with a low carbon number and double-bond content were the most positively associated with the risk of CVD [20]; the PREDIMED trial also found that glycerides with a stearic acyl chain were positively associated with the risk of CVD and that highly unsaturated CEs were inversely associated with it [32,33,34]. However, neither study adjusted for HDL-c and LDL-c. Thereafter, the WHI study identified TAGs and DAGs (diglycerides) with SFA and MUFA chains strongly associated with a high risk of CHD and showed that the associations were attenuated to null after adjusting for HDL-c and LDL-c [25]. Interestingly, even after adjusting for these lipid biomarkers, the MDC study and ADVANCE trial showed that TAG (48:1), TAG (48:2), TAG (48:3), TAG (50:3), TAG (50:4), and TAG (56:6) were inversely associated with the risk of CVD [19,24].

Fifth, as to findings of studies measuring lipids within lipoprotein subclasses using NMR, the FINRISK study and China Kadoorie Biobank study found that concentrations of total lipids, triglycerides, and cholesterol in very low density lipoproteins (VLDL), intermediate-density lipoproteins IDL, and LDL were positively associated with the risk of CVD, and concentrations of total lipids and cholesterol in HDL were inversely associated with it [21,22]. However, the FINRISK study further adjusted for total cholesterol and HDL-c and found that these associations were attenuated to null [21].

### 3.2. Potential Mechanisms of the Association of Ceramides and Fatty Acyl Chains with the Risk of CVD 

Ceramides have been incorporated for CVD risk prediction. The Finnish cohort showed that the ratio of two ceramides (18:1/16:0, 18:1/24:0) improved CHD prediction by 9% [31]. This ratio has been applied to clinical practice at the Mayo Clinic, representing the first lipidomic biomarker for CVD and demonstrating the potential of identifying novel lipid biomarkers using lipidomics [35]. Ceramides can be synthesized de novo by ceramide synthase (CerS). CerS N-acylates sphinganine and sphingosine, coupling them to a long-chain fatty acid to form ceramides. N-acylation of sphinganine (SA) forms dihydroceramide, which is then subsequently converted to Cer by dihydroceramide desaturase 1 (DES1), whereas N-acylation of sphingosine forms Cer directly. There are six CerS isoenzymes which differ in their acyl chain specificity. The altered ratio of ceramides 18:1/16:0 and 18:1/24:0 might be due to the fact that the activity or expression of certain CerS isoforms is changed in CVD. Moreover, ceramides as sphingolipids are crucial to membrane stability and have been shown to act as second messengers in membrane signaling pathways during inflammation [36]. Early study in humans showed that ceramides, particularly lactosylceramide and glucosylceramide, accumulate in atherosclerotic plaques and accelerate the development of atherosclerosis [37]. Animal studies have shown that the inhibition of ceramide synthesis significantly decreased the areas of atherosclerotic lesions, and enzymes in the ceramide synthetic pathway might serve as potential drug targets [38,39]. 

Our review showed that SFA and MUFA chains were associated with a high risk of CVD, while PUFA chains were inversely associated with it. This is consistent with findings from previous fatty acid biomarker research. Large cohort studies have found that plasma long-chain MUFA were associated with a high risk of CVD [40], while plasma omega-3 and omega-6 PUFA levels were associated with a low risks of CVD [41,42]. All three of these fatty acids can be obtained from diet, and PUFA have the strongest correlation between dietary intake and plasma levels. In fact, the correlations were 0.44 for PUFA, 0.24 for SFA, and 0.05 for MUFA [43]. Dietary PUFAs are obtained mainly from vegetable oils, nuts, seeds, and fish. Numerous studies have consistently shown that a healthful diet enriched in vegetables, nuts, and fish is associated with lower risks of CVD and CVD mortality compared with a diet poor in these nutrients [44,45,46]. The mechanism by which PUFAs lower the risk of CVD appears to be mainly mediated by changes in the concentrations of lipid biomarkers including HDL-c, LDL-c, and triglycerides [47]. One clinical trial demonstrated that higher intakes of PUFAs resulted in decreased LDL and increased HDL [48]. In addition to diet, SFAs and MUFAs can be produced internally through de novo lipogenesis. When energy is in excess, carbohydrates are converted into SFAs and MUFAs that are esterified into TAGs for storage in the liver and adipose tissue. Excess energy in the body results in obesity, which is a main risk factor for cardiometabolic diseases including CVD. This mechanism may partially explain why the associations of SFA, MUFA, and PUFA chains in TAGs and CEs with the risk of CVD are eliminated after additionally adjusting for total and HDL cholesterol.

## 4. Lipidome of Lipoproteins

Several studies have examined the lipid components of lipoproteins among healthy participants [49,50,51,52]. As summarized in Table 2, the lipid composition of HDLs, LDLs, and VLDLs varies. VLDLs are mainly composed of TAGs, while LDLs and HDLs are primarily composed of CEs and phospholipids. Compared to LDLs, HDLs contain higher percentages of phospholipids, particularly PC and LPC. The fatty acid composition of phospholipids, TAGs, and cholesterol seems comparable across HDLs, LDLs, and VLDLs. Given that the total lipid content differs across lipoproteins, a previous study has determined the distribution of the concentrations of lipid species across lipoproteins [50]. LPC in total plasma is predominantly contained in HDL particles; PC is mainly contained in HDL and LDL particles; and TAGs and DAGs are contained in all four lipoproteins. As for fatty acyl chains, PUFA chains are primarily contained in HDLs and LDLs, while MUFA and SFA chains are more likely to be distributed across VLDLs and chylomicrons. For ceramides, one study showed that gangliosides, which are glycosphingolipids, are primarily transported by LDL s(66%), followed by HDLs (25%) and VLDLs (7%) [53]. 

## 5. Revisiting Previous Findings on Lipidomics of Cardiovascular Disease

Given that ceramides are primarily contained in LDLs, the positive association between ceramide and risk of CVD may be due to the influence of ceramide on the function of LDLs. In fact, one previous study has shown that LDLs extracted from human atherosclerosis lesions are highly enriched in ceramide [54], and one animal study found that the decrease in size of atherosclerotic lesions might be due to inhibitors of ceramide synthesis that lower plasma LDL-c [39]. Overall, these findings indicate that ceramides might play an important role in LDL dysfunction and aggregation in atherosclerotic lesions, which needs further validation and investigation.

Our review showed that after adjusting for HDL-c and LDL-c, only the associations of PC with CVD outcomes remained, indicating that PCs are potential new biomarkers of CVD independent of total cholesterol and HDL-c. The phospholipids are of critical importance in maintaining the biological functions of HDL. Enrichment in phospholipid within HDL particles can increases HDL integrity and stability and prevent HDL clearance from plasma [55], while decreased levels of phospholipids in HDL have been reported to impair cholesterol efflux and decrease the CVD protective effects of HDLs [56]. Mainly contained in HDLs, LPC is produced upon hydrolysis of PC in acute and chronic inflammatory processes [57]. Studies have shown that the hydrolysis of PCs promotes the development of atherosclerosis and is associated with a high risk of CVD [58]. In the near future, it is worth further investigating why PCs with SFA/MUFA and PUFA chains are associated with CVD risk in opposite directions and whether the PC content affects the function of LDLs. 

As to SFAs and MUFAs esterified with TAGs and CEs, positive associations with the risk of CVD were found; however, the associations were attenuated to null after adjusting for HDL-c and LDL-c. The reason might be that the positive associations were mainly mediated through HDL-c, LDL-c, and TGAs [47]. Similarly, the null association between lipids in lipoproteins measured using NMR and risk of CVD after adjusting for HDL-c and LDL-c may be due to the high correlation of the levels of total lipids, TAGs, and CEs within lipoproteins with HDL-c or LDL-c intotal plasma. 

Overall, lipidomics of lipoproteins will enhance the mechanistic understanding of how lipids are associated with the risk of CVD. Given that a considerable number of lipids correlate with lipid biomarkers including HDL-c, LDL-c, and TAGs and that lipoproteins present distinct compositions of lipids, lipidomics of lipoproteins might better control the confounding effects of HDL-c and LDL-c and allow the identification of novel lipid biomarkers of CVD. 

## 6. Summary of Recent Studies on Lipidomics of Lipoproteins and CVD Outcomes

Relatively few studies have examined the lipidomics of lipoproteins and CVD outcomes; the findings are summarized in Table 3. In total, we identified eight studies using NMR or LC–MS for lipidomic analyses, all based on a relatively small sample size (number of participants ranging from 20 to 159). Most of the studies were conducted in hospital settings using cross-sectional or case–control designs [59,60,61,62,63,64,65]. Three studies conducted lipidomics of HDL and non-HDL particles [51,52,64], while five studies specifically focused on the lipidome of HDLs [61,62,63,65,66]. Of the five studies, two studies examined varieties in HDL lipid content, comparing participants with high HDL-c to those with low HDL-c [61,66], and the other studies focused on the lipidome of lipoproteins and CHD diagnosis and progression [51,52,62,63,64,65]. Only one study included LDL-c as a covariate [62], and only three studies corrected for multiple testing [63,64,65]. 

### 6.1. Findings and Potential Mechanisms

Based on the eight studies, higher TAGs, lower PUFAs, lower phospholipids, and lower sphingomyelin (SM) in HDLs may be associated with higher risks of CHD and type 2 diabetes (T2D), although it is unknown whether the associations are confounded by HDL-c. As consistently shown in two studies, a lower HDL-c, a risk factor of CHD, was associated with higher TAGs, lower PUFAs, lower LPC, and lower SM in HDLs [51,52]. Another study found that, compared to participants without CHD or at the initial stages of CHD, participants with severe CHD had higher levels of SFA chains in both HDL and non-HDL particles, lower levels of PC and SM in HDL particles, and lower levels of PUFA in non-HDL particles [60]. Additionally, two studies showed that lower levels of PC-plasmalogens were associated with a higher risk of CHD, particularly acute CHD [63,64]. Plasmalogens contain a vinyl ether-linked alkyl chain at the sn-1 position of the glycerol backbone and a polyunsaturated fatty acyl chain at the sn-2 position. Plasmalogens play an important role in reverse cholesterol transport, and plasmalogen-deficient cells showed impairment in cholesterol transport from the cell surface to the endoplasmic reticulum [67,68]. Furthermore, plasmalogens have been shown to prevent the oxidation of cholesterol in phospholipid bilayers [69].

### 6.2. Limitations

Previous studies found that differences in HDL lipid composition were associated with CVD outcomes, showing the promising application of lipoproteins’ lipidomics in CVD prevention and prediction. However, the number of studies remains limited, and the available studies included very small numbers of participants. The quality of these studies is also a concern, given that they used case–control or cross-sectional designs in hospital settings, and most of them did not adjust for HDL-c or LDL-c or correct for multiple testing. These limitations show the importance of conducting high-quality epidemiological studies to better understand how lipids within lipoproteins are associated with CVD outcomes. 

## 7. Conclusions

In conclusion, by summarizing previous publications on lipidomics and CVD outcomes, our review shows that lipidomics of lipoproteins to identify lipid association with CVD risk is an important research direction. Lipidomics will allow us to better understand how the identified lipid metabolites are associated with the risk of CVD and provide the opportunity to identify novel lipid biomarkers of CVD in HDLs and LDLs independent of HDL-c and LDL-c. The well-developed lipidomic techniques show the feasibility of this exciting research direction, and the lack of high-quality studies indicates that well-designed cohort studies are needed. 

## Figures and Tables

**Table 1 metabolites-10-00163-t001:** Main findings of previous lipidomic studies on the association of lipids with cardiovascular disease (CVD).

Main Findings	Findings Independent of Total and HDL Cholesterol	Studies
Ceramides (d18:1/16:0, d18:1/18:0, and d18:1/24:1) were positively associated with risk of CVD outcomes.	Yes	The findings were first documented in the LURIC study [23] and replicated in the ATHEROREMO-IVUS study [28,29]. FINRISK 2002 study, Corogene study, PREDIMED trial, ADVANCE study, and LIPID study consistently confirmed the findings [24,26,30,31,32].
Phosphatidylcholines (PC) with saturated (SFA) and monounsaturated (MUFA) fatty acyl chains were positively associated with risk of CVD outcomes, while PCs with polyunsaturated fatty acyl chains (PUFA) were inversely associated.	Yes	The findings were observed in the LURIC study, WHI, PREDIMED trial, ADVANCE trial, and LIPID study [23,24,26,32,33,34], and the associations persisted after adjusting for HDL-c and LDL-c.
Lysophosphatidylcholines (LPC) may be positively associated with risks of CVD outcomes.	Yes	The findings were observed in the LIPID and ADVANCE trials [24,26]. However, inverse associations of LPCs with risk of CVD outcomes were found in the MDC study, TwinGene study, and LURIC study [19,23,27]. No associations for LPCs with risk of coronary heart disease (CHD) were documented in the WHI [25].
Triglycerides (TAGs) and cholesterol esters (CE) with SFA and MUFA chains were positively associated with risk of CVD outcomes, while CEs with PUFA chain were inversely associated.	No	The findings were observed in the Bruneck study, PREDIMED trial, and WHI study [20,25,32,33,34]. However, after further adjusting for HDL-c and LDL-c, the WHI study found that these associations were eliminated.
Measured by NMR, concentrations of total lipids, TAG, and CE in VLDL, IDL, LDL were positively associated with risk of CVD.	No	The findings were observed in the FINRISK study and China Kadoorie Biobank study [21,22]. However, the FINRISK study further adjusted for HDL-c and LDL-c and found that these associations were eliminated [21].

LURIC: LUdwigshafen RIsk and Cardiovascular Health; ADVANCE trial: the Action in Diabetes and Vascular Disease: Preterax and Diamicron MR Controlled Evaluation (ADVANCE) trial; PREDIMED trial: The Prevención con Dieta Mediterránea (PREDIMED) trial; WHI: Women’s Health Initiative; NMR: Nuclear magnetic resonance; VLDL: Very low density lipoprotein; IDL: Intermediate-density lipoprotein; LDL: Low-density lipoprotein; HDL: High-density lipoprotein.

**Table 2 metabolites-10-00163-t002:** Lipid composition of VLDLs, LDLs, and HDLs [50,52].

Lipid Species *	VLDL (%)	LDL (%)	HDL (%)
Phospholipids (PL)	11.1	12	37.4
LPC	2.3	0.4	3
PC	8.5	11.6	31.5
*PL-SFA*	49	50	46
*PL-MUFA*	12	12	12
*PL-PUFA*	34	32	37
Triglycerides (TAGs)	59	10	6.3
*TAG-SFA*	30	26	27
*TAG-MUFA*	45	47	44
*TAG-PUFA*	18	21	24
Cholesterol esters (CEs)	21.6	74.5	54.7
*CE-SFA*	13	12	12
*CE-MUFA*	26	22	22
*CE-PUFA*	58	67	61

* For lipid species, percentages indicate weight of each lipid species over weight of total lipids within each lipoprotein. For fatty acyl composition, percentages indicate weight of fatty acyl chain over weight of total fatty acyl chain within the lipid species of each lipoprotein.

**Table 3 metabolites-10-00163-t003:** A summary of the studies on lipidomics of lipoproteins reporting cardiovascular outcomes.

Author, Country, Publication Year	Study Design	Number of Participants	Platform	Outcome	Findings	Adjusting for HDL-c, LDL-c, or TAGs
Yetukuri et al. [66], Finland, 2010	Cross-sectional study in a survey	47 participants (24 with low HDL-c and 23 with high HDL-c)	Lipidomics of HDLs measured by LC–MS, with 307 lipids identified.	Lipidome of HDL	Higher HDL-c was associated with lower TAGs (48:0, 48:1, 54:3), higher LPC (22:6, 18:1, 18:0), and higher SM (d18:1/16:0, d18:1/22:0, d18:1/24:1) in HDL. No correction for multiple testing.	-
Kostara et al. [61], Greece, 2017	Cross-sectional study in hospital setting	60 healthy participants (20 with low HDL-c, 20 with normal HDL-c, and 20 with high HDL-c)	Lipidomics of HDLs measured by NMR	Lipidome of HDL	Higher HDL-c was associated with lower TAGs, higher PUFA chain, and higher SM in HDL. No correction for multiple testing.	-
Kostara et al. [59,60], Greece, 2009, 2014	Case–control study in hospital setting	99 CHD cases (30 cases with mild CHD, 29 with moderate CHD, and 40 with severe CHD) and 60 controls	Lipidomics of HDL and non-HDL particles measured by NMR	Progression of CHD	Participants with more severe CHD presented higher levels of SFA chains in HDL and non-HDL particles, lower levels of PC and SM in HDL particles, and lower levels of PUFA chains in lipids in non-HDL particles. No correction for multiple testing.	-
Morgantini et al. [62], Italy, 2014	Case–control study in hospital setting	80 participants without CVD and 38 CVD cases	HETEs and HODEs in HDLs measured by LC–MS	CVD	HETEs (15-HETE, 12-HETE, 5-HETE) and HODEs (13-HODE, 9-HODE) content in HDLy were significantly higher in CVD cases in comparison to participants without CVD. No correction for multiple testing.	LDL-c
Sutter et al. [63], Switzerland, 2015	Case–control study in hospital setting	22 healthy participants and 45 CHD cases	49 PCs, LPCs, and SMs, and 3 S1P in HDLs	CHD	Levels of PC-derived plasmalogens in HDLs (PC33:3, PC35:2, PC35:5) were significantly lower in CHD cases compared to controls.	-
Meikle et al. [64], Australia, 2019	Case–control study in hospital setting	47 participants with acute CHD and 83 with stable CHD	Lipidomics of HDLs and LDL by LC–MS	Subtypes of CHD	Level of lysophospholipids and plasmalogens in HDLs were significantly lower among acute CHD participants in comparison to patients with stable CHD.	Statin use
Cardner et al. [65], Switzerland, 2020	Case–control study in hospital setting	51 healthy subjects and 98 cases with T2D or CHD	Lipidomics of HDsL by ESI–MS	CHD or T2D	T2D or CHD cases presented higher PEs (38:5, 38:6, 40:7) and lower PIs (36:2, 34:2), PCs (36:2, 34:2), and CE 18:2. T2D cases had significantly lower levels of PCs (O-34:2, O-34:3, O-36:2, O-36:3), LPCs (18:2, 18:1, 18:0), and SMs (42:3 and 36:3).	-

NMR: Nuclear magnetic resonance; LC–MS: Liquid chromatography–mass spectrometry; ESI-MS: Electrospray ionization–tandem mass spectrometry SM: Sphingomyelin; HETE: Hydroeicosatetraenoic acid; HODE: Hydroxyoctadecadienoic acids; S1P: Sphingosine-1-phosphate; PE: Phosphatidylethanolamines; PI: Phosphatidylinositols.

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
