# Peer review of "A Review of Lipidomics of Cardiovascular Disease Highlights the Importance of Isolating Lipoproteins"

_metabolites, 2020, doi:10.3390/metabo10040163_

Round 1

Reviewer 1 Report

The authors amendments make the paper much clearer to read. I have only a couple of very trivial amendments to suggest to improve clarity for the reader.

Section 2: the authors state that DCM/methanol or butanol/methanol is the most popular method. Other organic solvents including chloroform and MTBE are still widely used. While the methods referenced by the authors are widely used, either clear evidence should be provided that they are the "most popular" or change to simply "popular"

Section 2: the section on ion suppression suggests it is a detector deficiency, while the actual cause is reduced ionisation such that reduced numbers of ions are available to reach the detector. Please make the explanation clearer.

P6, first line: "PCs containing SFA and MUFA were positively associated with CVD outcomes" This sentence is slightly unclear as it is not clear what a CVD outcome is in this context (good outcome/bad outcome?). Please clarify the sentence e.g. "PCs containing SFA and MUFA were positively associated with a higher risk of CVD" or similar.

Author Response

The authors amendments make the paper much clearer to read. I have only a couple of very trivial amendments to suggest to improve clarity for the reader.

Section 2: the authors state that DCM/methanol or butanol/methanol is the most popular method. Other organic solvents including chloroform and MTBE are still widely used. While the methods referenced by the authors are widely used, either clear evidence should be provided that they are the "most popular" or change to simply "popular"

We have changed the word to “popular” (section 2.1, page 2).

Section 2: the section on ion suppression suggests it is a detector deficiency, while the actual cause is reduced ionisation such that reduced numbers of ions are available to reach the detector. Please make the explanation clearer.

We have revised the sentence as “another limitation of LC-MS is ion suppression, which is defined as reduced ionisation such that reduced numbers of ions are available to reach the detector.” (section 2.1, page 2).

P6, first line: "PCs containing SFA and MUFA were positively associated with CVD outcomes" This sentence is slightly unclear as it is not clear what a CVD outcome is in this context (good outcome/bad outcome?). Please clarify the sentence e.g. "PCs containing SFA and MUFA were positively associated with a higher risk of CVD" or similar.

We have revised the sentence as “PCs containing SFA and MUFA were associated with higher risk of CVD” (paragraph 1, page 6).

Reviewer 2 Report

I am satisfied with the corrections done by the authors and I think the new version of the manuscript has been improved.

I just have some comments, mainly format issues: 

In section 2.6, please unify the format of shot-gun lipidomics (authors previously wrote shotgun). The same in section 4.1, TG but TAG in the whole manuscript.

There is an extra-page (page 4).

Finally, unless I am mistaken, the abbreviation "CHD" is not defined in the text and in the abstract, you put "sphingomyelin" without the abbrevition SM. Moreover, in section 3, authors declared "We searched recent cohort-based studies", what do you mean with "recent"? Could you be more specific about the period?. I need to apologyze to the authors because these comments are new, but there are minor comments I think no difficult to address them. 

Author Response

I am satisfied with the corrections done by the authors and I think the new version of the manuscript has been improved.

I just have some comments, mainly format issues: 

In section 2.6, please unify the format of shot-gun lipidomics (authors previously wrote shotgun). The same in section 4.1, TG but TAG in the whole manuscript.

We have revised shot-gun into shotgun (section 2.3, page 3), and TG into TAG (section 4.1, page 8).

There is an extra-page (page 4).

I do not know how to delete the page without changing the format and look forward to the editor’s help on it.

Finally, unless I am mistaken, the abbreviation "CHD" is not defined in the text and in the abstract, you put "sphingomyelin" without the abbrevition SM. Moreover, in section 3, authors declared "We searched recent cohort-based studies", what do you mean with "recent"? Could you be more specific about the period?. I need to apologyze to the authors because these comments are new, but there are minor comments I think no difficult to address them. 

We now have spelt out CHD for coronary heart disease in the abstract (page 1) and section 3 (page 3), and SM for sphingomyelin in section 5.1 (page 8). We have deleted the word recent in section 3 (page 3).

Reviewer 3 Report

There is a minor incorrectness concerning CerS activity (section 3.2). CerS can N-acylate both Sphinganine (SA) and Sphingosine (SO). N-acylation of SA forms dihydroCeramid (SA) (which is then subsequently converted to Cer by DES1) whereas N-acylation of SO forms Cer directly. This should be revised.

All other concerns were met so far and I agree with the publication of the revised version.

Author Response

There is a minor incorrectness concerning CerS activity (section 3.2).

 CerS can N-acylate both Sphinganine (SA) and Sphingosine (SO). N-acylation of SA forms dihydroCeramid (SA) (which is then subsequently converted to Cer by DES1) whereas N-acylation of SO forms Cer directly. This should be revised.

As the reviewer suggested, we have revised the sentence as “CerS N-acylates coupling of sphinganine and sphingosine to a long-chain fatty acid to form ceramides. N-acylation of sphinganine forms dihydroCeramid (SA), which is then subsequently converted to Cer by dihydroceramide desaturase 1 (DES1), whereas N-acylation of sphingosine forms Cer directly.”(section 3.2, page 6).

This manuscript is a resubmission of an earlier submission. The following is a list of the peer review reports and author responses from that submission.

Round 1

Reviewer 1 Report

The authors have written an interesting piece which appears to be half review, half meta-analysis of the importance of lipid classes in cardiac disease. I would recommend that they either reassess and write it as a full and comprehesive review of the field, or make it a full meta analysis paper with all the appropriate detail provided. I also think that it may be better placed in a journal with a more epidemiological scope.

The results and conclusions were interesting, but I have the following concerns;

General:

The sections on the technical aspects of measurement were poorly described, and occasionally misleading.

The acronyms were not explained

This is described as a review but the article here seems to go into meta analysis. This is difficult to review since the exact methods were not appropriately described.

Specifically section 2.1:

You describe DCM/Methanol as the most popular method for lipoprotein analysis based on a single review article that is now 6 years old. There are other methods being widely used which have not been discussed.

LC-MS cannot identifiy structural isomers because there separation times are too similar i.e. it is not about molecules competing for ionization as suggested here.

LC-structurally-based ion mobility spectrometry is misleading as a term as it suggests the LC is structurally based. Since there is size exclusion chromatography, this is confusing for the reader. I would suggest LC-IMS-MS which is the more standard term.

Section 2.2: ESI does not extend the range of the mass analyzer as you state.

Section 3: This is described as a review but the article here seems to go into meta analysis.You mention three studies (FINRISK, Corogene and Predimed that focus on a very narrow field of lipoproteins as follow up studies but then mention in table 3 that no FDR correction was applied. How many lipids were being assessed? You report 4. Do you believe FDR correction would make a significant difference to the results?

In the entire of this section it was not clear to me what you were reporting from previous studies and what you had reanalysed yourself.

You referred repeatedly to „standard lipids“ which is a term I am unfamiliar with and requires further explanation.

You need to describe your full analysis methods and be clear when you are describing free lipids and lipoprotein lipids.

Author Response

Reviewer 1

The authors have written an interesting piece which appears to be half review, half meta-analysis of the importance of lipid classes in cardiac disease. I would recommend that they either reassess and write it as a full and comprehensive review of the field, or make it a full meta-analysis paper with all the appropriate detail provided. I also think that it may be better placed in a journal with a more epidemiological scope.

In this paper, although we listed the findings from each study as a table, we did not pool the results across studies. Thus, this is not a meta-analysis paper. Instead, we quantitatively summarized results from all studies, and this is a comprehensive systematic review on lipidomics of CVD.

The results and conclusions were interesting, but I have the following concerns;

General:

The sections on the technical aspects of measurement were poorly described, and occasionally misleading.

We have made revisions to the technical section, as the reviewer suggested.

The acronyms were not explained

We have added full names of acronyms.

This is described as a review but the article here seems to go into meta analysis. This is difficult to review since the exact methods were not appropriately described.

As aforementioned, our study is not a meta-analysis, but a comprehensive review. We extracted main findings from each study, and quantitatively summarized the results.

Specifically section 2.1:

You describe DCM/Methanol as the most popular method for lipoprotein analysis based on a single review article that is now 6 years old. There are other methods being widely used which have not been discussed.

We have included other liquid-liquid extraction methods in the main text and additionally included two recent studies as references.

LC-MS cannot identify structural isomers because their separation times are too similar i.e. it is not about molecules competing for ionization as suggested here.

We have made revisions regarding separation time as the reviewer suggested (page 5).

LC-structurally-based ion mobility spectrometry is misleading as a term as it suggests the LC is structurally based. Since there is size exclusion chromatography, this is confusing for the reader. I would suggest LC-IMS-MS which is the more standard term.

Based on the comments from reviewers 1 and 3, we have removed the sentence on LC-IMS-MS.

Section 2.2: ESI does not extend the range of the mass analyzer as you state.

We have removed “extend the range of the mass analyzer” as the reviewer suggested.

Section 3: This is described as a review but the article here seems to go into meta analysis.You mention three studies (FINRISK, Corogene and Predimed that focus on a very narrow field of lipoproteins as follow up studies but then mention in table 3 that no FDR correction was applied. How many lipids were being assessed? You report 4. Do you believe FDR correction would make a significant difference to the results?

FINRISK, Corogene and Predimed only included four lipids, and we have described in the text that “the FINRISK 2002, the Corogene study, and the PREDIMED Trial specifically focused on these four ceramides and consistently found that Cer(d18:1/16:0), Cer(d18:1/18:0), and Cer(d18:1/24:1) were positively associated with CVD outcomes”. These findings were listed in supplemental table 1 and summarized in table 1.

In table 3, we listed findings on lipidomics of lipoproteins. These studies used different populations other than FINRISK, Corogene and Predimed and included many lipid metabolites. As we addressed in the text, these studies did not correct for multiple testing, which is a main limitation.

In the entire of this section it was not clear to me what you were reporting from previous studies and what you had reanalyzed yourself.

This is a comprehensive review of previous studies. We quantitively described findings from previous studies. We did not reanalyze any data ourselves.

You referred repeatedly to „standard lipids“ which is a term I am unfamiliar with and requires further explanation.

For improved clarity, we have replaced standard lipids with HDL-c, LDL-c, and TAG throughout the manuscript.

You need to describe your full analysis methods and be clear when you are describing free lipids and lipoprotein lipids.

This is a review paper and we did not analyze data ourselves. As the reviewer suggested, we have clarified the description of free lipids and lipoprotein lipids.

Reviewer 2 Report

The authors present a really interesting review focused on the cohort studies which used lipidomics to study the progression of CVD. I especially like the way that the authors summarized the previous studies by extracting intengrative conclusions of the considered works, which can facilitate the work to other authors interested in this area of the knowlegde.

I have just a question regarding the figure 1, do the author have the persimisson for publising that figure?. In fact, in my opinion the figure is not really necessary (just cite it in the text along with the reference) because it is extremmenly difficult to see it properly.

Maybe, the authors should include the small number of studies as a limitation. 

Author Response

Reviewer 2

The authors present a really interesting review focused on the cohort studies which used lipidomics to study the progression of CVD. I especially like the way that the authors summarized the previous studies by extracting integrative conclusions of the considered works, which can facilitate the work to other authors interested in this area of the knowledge.

We appreciate the reviewer’s positive comments on our review.

I have just a question regarding the figure 1, do the author have the permission for publishing that figure? In fact, in my opinion the figure is not really necessary (just cite it in the text along with the reference) because it is extremely difficult to see it properly.

As the reviewer suggested, we have removed Figure 1 from the manuscript and cited it as a reference.

Maybe, the authors should include the small number of studies as a limitation. 

As the reviewer suggested, we have included small number of studies as a limitation (page 14).

Reviewer 3 Report

The review provides a good overview on the major lipidomics studies performed in the context of CVD. However, concerning the cited literature the manuscript is a bit gappy. Among others, the authors may consider to include and discuss these recent publications:

Cardner et al JCI Insight. 2020 Jan 16

Jayawardana et al, JCI Insight. 2019 Jul 11;4(13)

Meikle et al J Am Heart Assoc. 2019 Jun 4;8(11)

The description of the lipidomics techniques is partly erroneous. Structural and positional isomers are (often) separated by LC-MS but can’t normally be resolved with shotgun MS (exactly the opposite as described). The authors should mention the two mostly used LC methods in lipidomics (normal vs reverse phase). The problem of ion suppression in LC-MS and relative versus absolute quantification should be discussed in respect to the described MS method (LC-MS and Shotgun Lipidomics). This section might profit from being reviewed by somebody with hands-on experiences in mass spectrometry.

The disadvantages of NMR based lipidomics should be discussed (low sensitivity, relatively long measurement times, little sample throughput)

The authors might mention reduced plasmalogens as CVD risk markers (Suter et al  Atherosclerosis. 2015 Aug;241(2):539-46 ; Paul S, Prog Lipid Res. 2019 Apr;74:186-195)

What data are used as base for Fig 1? Please add reference

Table 2 – sphingolipids (in particular sphingomyelin) are also constituent of lipoproteins and should be added.

To me the term "standard lipids" for HDL, LDL etc is too ambiguous. HDL and LDL are lipoproteins, which by itself are composed of lipids and proteins. I'd also suggest to be more specific on the type of adjustments made for the described studies. The adjustments are also important for the reader to know

On the mechanistic side, the authors might also refer to the metabolic base for the difference in the CVD associated Cer species ((d18:1/16:0, d18:1/18:0 and d18:1/24:1). These species are formed by a group of six ceramide synthase (CerS) isoenzymes which differ in their acyl chain specificity. The altered ratio of these ceramides in CVD indicates that the activity/expression of certain CerS isoforms is changed in CVD.

Author Response

Reviewer 3

The review provides a good overview on the major lipidomics studies performed in the context of CVD. However, concerning the cited literature the manuscript is a bit gappy. Among others, the authors may consider to include and discuss these recent publications:

Cardner et al JCI Insight. 2020 Jan 16

Jayawardana et al, JCI Insight. 2019 Jul 11;4(13)

Meikle et al J Am Heart Assoc. 2019 Jun 4;8(11)

As the reviewer suggested, we have included these studies in Table 2, and have discussed them in the main text (See page 14).

The description of the lipidomics techniques is partly erroneous. Structural and positional isomers are (often) separated by LC-MS but can’t normally be resolved with shotgun MS (exactly the opposite as described).

In our manuscript, we have mentioned that “distinguishing lipid isomers is difficult using shotgun lipidomics, due to low efficiency of separating aggregated lipids.” Based on the comments from reviewers 1 and 3, we have removed the sentence about isomers using IMS-MS (page 5).

The authors should mention the two mostly used LC methods in lipidomics (normal vs reverse phase).

We have included “LC separation includes normal phase LC, with polar stationary phase and non-polar mobile phase, and reversed phase LC, with non-polar stationary phase and polar mobile phase.” (page 5)

The problem of ion suppression in LC-MS and relative versus absolute quantification should be discussed in respect to the described MS method (LC-MS and Shotgun Lipidomics). This section might profit from being reviewed by somebody with hands-on experiences in mass spectrometry.

We have included ion suppression as a limitation for LC-MS and shotgun lipidomics (page 6).

The disadvantages of NMR based lipidomics should be discussed (low sensitivity, relatively long measurement times, little sample throughput).

We have included the disadvantages of NMR in the main text (page 7).

The authors might mention reduced plasmalogens as CVD risk markers (Suter et al Atherosclerosis. 2015 Aug;241(2):539-46; Paul S, Prog Lipid Res. 2019 Apr;74:186-195)

We have included the paper by Suter et al in Table 2 and the potential mechanisms of reduced plasmalogens as biomarkers by Paul et al in the main text (page 14).

What data are used as base for Fig 1? Please add reference

We have included a reference in the main text “Christinat N et al. Journal of proteome research. 2017;16:2947-2953.”

Table 2 – sphingolipids (in particular sphingomyelin) are also constituent of lipoproteins and should be added.

We agree with the reviewer that sphingolipids are an important constituent of lipoproteins. However, it is difficult to find any literature that reported composition of sphingolipids in lipoproteins. We found composition of a subspecies of ceramides and have included in the text:  “For ceramides, one study showed that gangliosides, a glycosphingolipid, are primarily transported by LDL (66%), followed by HDL (25%) and VLDL (7%).” (page 12).

To me the term "standard lipids" for HDL, LDL etc is too ambiguous. HDL and LDL are lipoproteins, which by itself are composed of lipids and proteins.

As described above, for clarity we have replaced standard lipids with HDL-c, LDL-c, and TAG throughout the manuscript.

I'd also suggest to be more specific on the type of adjustments made for the described studies. The adjustments are also important for the reader to know.

As the reviewer suggested, we have included a column on covariates adjusted for in Supplemental table 1.

On the mechanistic side, the authors might also refer to the metabolic base for the difference in the CVD associated Cer species ((d18:1/16:0, d18:1/18:0 and d18:1/24:1). These species are formed by a group of six ceramide synthase (CerS) isoenzymes which differ in their acyl chain specificity. The altered ratio of these ceramides in CVD indicates that the activity/expression of certain CerS isoforms is changed in CVD.

As the reviewer suggested, we have included this potential mechanism for Cer species in the main text (page 10).